# Survival outcomes of breast cancer patients with recurrence after surgery according to period and subtype

**Young-jin Lee**[ID], **Tae-Kyung Yoo**[⊛], **Jisun Kim**[⊛], **Il Yong Chung**[⊛], **Beom Seok Ko**[⊛], **Hee Jeong Kim**[⊛], **Jong Won Lee**[⊛], **Byung Ho Son**[⊛], **Sei-Hyun Ahn**[⊛], **Sae Byul Lee**[ID]*

Department of Surgery, Division of Breast Surgery, Asan Medical Center, University of Ulsan College of Medicine, Seoul, Republic of Korea

⊛ These authors contributed equally to this work.
* newstar153@hanmail.net

**Data Availability Statement:** All relevant data are within the paper and its Supporting Information files.

## Abstract

### Purpose

To analyze and compare the survival rates of recurrent breast cancer patients in Korea between two periods (period I: 2000–2007; period II: 2008–2013) and to identify the factors associated with outcomes and changes over time in the duration of survival after recurrence.

### Methods

We retrospectively analyzed 2,407 patients who had recurrent breast cancer with treated between January 2000 and December 2013 and divided them into two periods according to the year of recurrence. We reviewed the age at diagnosis, clinical manifestations, pathology report, surgical methods, types of adjuvant treatment, type of recurrence, and follow-up period.

### Results

The median follow-up was 30.6 months (range, 0–223.4) from the time of relapse, and the median survival time was 42.3 months. Survival after recurrence (SAR) significantly improved from 38.0 months in period I to 49.7 months in period II ($p < 0.001$). In the analysis performed according to the hormone receptor and HER2 status subtypes, all subtypes except the triple-negative subtype showed higher SAR in period II than period I. Age at diagnosis, tumor stage, and treatment after recurrence were significantly correlated with survival outcomes.

### Conclusion

The survival outcomes of Korean patients with breast cancer after the first recurrence have improved in Korea. Such improvements may be attributed to advances in treatment.

**Funding:** The authors received no specific funding for this work.

**Competing interests:** The authors have declared that no competing interests exist.

## Introduction

Breast cancer is the most common cancer in women globally. Recently, breast cancer has become the most common cancer in Korean women as well [1,2]. More than 80% of breast cancers are detected and treated in a curative state at the time of diagnosis [3]. Some patients with breast cancer experience recurrence after a certain period after treatment, most of whom eventually die from disease recurrence. Therefore, elucidating the factors influencing the recurrence of cancer would be helpful in estimating the treatment outcomes and predicting the prognosis.

Several studies have identified some of these prognostic factors [4,5], including tumor size, nodal stage, hormone-receptor status, tumor grade, and adjuvant therapy [6,7]. In contrast, there have been only few studies on prognostic factors affecting survival after recurrence (SAR), such as the hormonal status of the tumor, extent of metastasis, interval to recurrence, and the presence of visceral metastasis [8,9]. However, the knowledge on these prognostic factors is relatively poor and requires further investigation. In addition, the prognosis of relapsed patients is expected to have been greatly affected by the recently developed adjuvant treatment. Therefore, it is necessary to analyze the impact of the development of adjuvant therapy over time for each subtype on the prognosis of relapsed patients.

In this study, we retrospectively analyzed 2,407 patients with recurrent breast cancer. To analyze the effect of variations in treatments over time, we compared the characteristics and survival of patients according to the year of recurrence and categorized them into two periods to identify the factors associated with overall and post-recurrence survival.

## Methods

### Patients and clinical data

We retrieved the medical records of 17,776 female patients who were diagnosed with breast cancer and underwent breast surgery between January 2000 and December 2013 at Asan Medical Center (Seoul, Republic of Korea). And the participants were recruited into our database from 2017 to 2021. Among them, we selected 2,407 patients who experienced recurrence before December 31, 2020. We conducted an analysis in April 2021. The study period was divided into period I and period II according to significant changes made in the practice of adjuvant therapy.

All information about the patients and diseases was retrieved from the retrospectively collected database, including age; clinical manifestations; clinical and pathological staging according to the American Joint Committee on Cancer classification; pathological data; surgical methods; types of adjuvant therapy received during the first treatment; types of post-relapse adjuvant therapy, which marked the "after recurrence," type; and follow-up period. Overall survival (OS) was defined as the time from surgery to death/last follow up. SAR was defined as the time from recurrence to death/last follow up by referring to the Korean registry cause-of-death records.

### Pathological data

The tumor size, number of axillary lymph node metastases, estrogen receptor (ER) status, progesterone receptor (PR) status, and human epidermal growth factor receptor-2 (HER2) status were evaluated at the Department of Pathology at Asan Medical Center. The statuses of ER, PR, and HER2 were determined using immunohistochemical analysis. ER and PR statuses were considered to be positive when the tumor cells showed more than 1% positivity. HER2

overexpression grades 0, 1+, and 2+ were considered to be negative. Cases rated 2+ were evaluated using fluorescence in situ hybridization and those rated 3+ were considered positive.

### Follow-up routine

All patients received standard combination treatment, including surgery and adjuvant treatment, when the disease first presented. After adjuvant therapy, all patients were regularly followed every 6 months for the first 60 months, including clinical examinations, laboratory tests (include CA15-3), mammograms, ultrasonography, and chest X-rays. In the sixth year, follow-up was carried out annually until the first recurrence of the disease. Computed tomography, magnetic resonance imaging, or 18F-fluorodeoxyglucose positron emission tomography/computed tomography scans were performed when patients complained of symptoms suggestive of tumor recurrence.

### Statistical analysis

Data analysis was performed using IBM SPSS Statistics for Windows, version 21.0 (IBM Corp., Armonk, NY, USA). Linear regression analysis and chi-squared tests were used to determine the trend of each parameter over time. Survival curves were generated using the Kaplan–Meier method, and the significance of survival differences among selected variables was verified using the log-rank test. A univariate Cox regression analysis was used to estimate the hazard ratios. A multivariate Cox regression analysis with a backward elimination method was used to estimate the hazard ratios and p values and to identify independent prognostic factors. The unknown groups of each variables were removed prior to Cox analysis. All reported p-values were two-sided, and p values < 0.05 were considered statistically significant.

### Ethical approval

All procedures performed in studies involving human participants were in accordance with the ethical standards of the institutional and/or national research committee and with the 1964 Helsinki Declaration and its later amendments or comparable ethical standardsData analysis.

This study was reviewed and approved by the Institutional Review Board of Asan Medical Center (approval #2017–1341). The authors have access to the data of individual participants who have been anonymized. Informed consent was waived because the study was based on retrospective clinical data.

## Results

### Patient characteristics

The characteristics of 2,407 patients with breast cancer recurrence are shown in Table 1. The most common characteristics were as follows: age 35–50 at diagnosis (n = 1,320 [54.8%]); T2 stage disease (n = 1,123 [46.9%]); positive regional lymph node metastasis (n = 1,315 [54.6%]); histologic grade 3 (n = 1,050 [49.7%]); hormone receptor-positive and HER2-negative (n = 1,076 [46.3%]).

According to the year at diagnosis, the patients were categorized into period I (2000–2007; n = 1,257 [52.2%]) and period II (2008–2013; n = 1,150 [47.8%]) (Table 2). The distributions of T stage (p = 0.037) and nodal stage (p < 0.001) were significantly different between the two periods. T1 and N-positive breast cancer occupied large proportion of patients in period I. The following characteristics were more commonly observed in period II than in period I: hormone receptor-negative (p = 0.001), HER-2-negative (p = 0.008), no adjuvant radiotherapy for

**Table 1. Characteristics of total patients.**

| Factors | Patients (N = 2,407) | |
| --- | --- | --- |
| | **No.** | **%** |
| Age at diagnosis (y) | | |
| <35 | 312 | 13.0 |
| 35–50 | 1,320 | 54.8 |
| >50 | 775 | 32.2 |
| T stage | | |
| T1 | 816 | 34.1 |
| T2 | 1,123 | 46.9 |
| T3 | 238 | 9.9 |
| T4 | 108 | 4.5 |
| Tis | 110 | 4.6 |
| Unknown | 12 | |
| Nodal stage | | |
| Negative | 1,092 | 45.4 |
| Positive | 1,315 | 54.6 |
| Stage | | |
| Stage 0 | 108 | 4.5 |
| Stage I | 546 | 22.8 |
| Stage II | 1,037 | 43.3 |
| Stage III | 703 | 29.4 |
| Unknown | 13 | |
| Histologic grade | | |
| G1 | 46 | 2.2 |
| G2 | 1,015 | 48.1 |
| G3 | 1,050 | 49.7 |
| Unknown | 296 | |
| Nuclear grade | | |
| G1 | 45 | 2.2 |
| G2 | 959 | 47.9 |
| G3 | 999 | 49.9 |
| Unknown | 404 | |
| LVI | | |
| No | 1,095 | 55.3 |
| Yes | 886 | 44.7 |
| Unknown | 426 | |
| Hormone receptor status | | |
| Negative | 926 | 40.4 |
| Positive[a] | 1,368 | 59.6 |
| Unknown | 83 | |
| HER-2 (IHC) status | | |
| Negative | 1,657 | 71.3 |
| Positive[b] | 667 | 28.7 |
| Unknown | 83 | |
| Subtype | | |
| HR+/HER-2- | 1,076 | 46.3 |
| HR+HER-2+ | 292 | 12.6 |
| HR-/HER-2+ | 345 | 16.1 |

(*Continued*)

**Table 1.** (Continued)

| Factors | Patients (N = 2,407) | |
|---|---|---|
| | No. | % |
| HR-/HER-2- | 581 | 25.0 |
| Unknown | 83 | |
| Breast surgery | | |
| Breast conservation surgery | 989 | 41.1 |
| Mastectomy | 1,418 | 58.9 |
| Radiotherapy | | |
| Yes | 1,535 | 64.7 |
| No | 839 | 35.3 |
| Unknown | 33 | |
| Chemotherapy | | |
| Yes | 1,817 | 23.5 |
| No | 559 | 76.5 |
| Unknown | 31 | |
| Hormonal therapy | | |
| Yes | 1,425 | 60.4 |
| No | 935 | 39.6 |
| Unknown | 47 | |
| Chemotherapy agent | | |
| None | 553 | 26.7 |
| CMF | 52 | 2.5 |
| Anthracyclin-based | 672 | 32.5 |
| Taxane-based | 757 | 36.6 |
| Others | 36 | 1.7 |
| Unknown | 337 | |
| Hormonal therapy agent | | |
| None | 944 | 40.1 |
| AI | 191 | 8.1 |
| SERM | 1,061 | 45.0 |
| SERM+LHRH analog | 160 | 6.8 |
| Unknown | 51 | |

LVI, lymphovascular invasion; [a]estrogen receptor-positive or progesterone receptor-positive; HER-2, human epidermal growth factor receptor-2; IHC, immunohistochemistry; [b]IHC 3+; HR, Hormone receptor; CMF, cyclophosphamide, methotrexate, fluorouracil; AI, aromatase inhibitor; SERM, Selective estrogen receptor modulator; LHRH, luteinizing hormone-releasing hormone.

initial breast cancer (p < 0.001), treated with adjuvant chemotherapy (p = 0.001), and treated with hormonal therapy (p < 0.001) (Table 2).

## Recurrence

Table 3 shows the distribution of the first recurrence sites according to the study period. The proportion of systemic decreased with time from period I to period II: in period I, 302 (24.0%) patients had loco-regional recurrence and 955 (76.0%) had systemic recurrence; in period II, 404 (35.0%) patients had loco-regional recurrence and 746 (65.0%) had systemic recurrence. There were significant differences in the type of recurrence according to the time period (p < 0.001).

**Table 2. Clinicopathologic characteristics of patients according to the year of recurrence.**

| Factors | | 2000–2007 (N = 1,257) | 2008–2013 (N = 1,150) | *p-value* |
|---|---|---|---|---|
| | | N (%) | N (%) | |
| Age at diagnosis (y) | <35 | 169 (13.4) | 143 (12.4) | 0.40 |
| | 35–50 | 698 (55.5) | 622 (54.1) | |
| | >50 | 390 (31.0) | 385 (33.5) | |
| Stage | 0 | 51 (4.1) | 57 (5.0) | 0.40 |
| | I | 275 (21.9) | 271 (23.8) | |
| | II | 557 (44.3) | 480 (42.2) | |
| | III | 374 (29.8) | 329 (28.9) | |
| | Unknown | 0 | 13 | |
| T stage | T1 | 443 (35.2) | 373 (32.8) | 0.037 |
| | T2 | 590 (46.9) | 533 (46.8) | |
| | T3 | 107 (8.5) | 131 (11.5) | |
| | T4 | 65 (5.2) | 43 (3.8) | |
| | Tis | 52 (4.1) | 58 (5.1) | |
| | Unknown | 0 | 12 | |
| Nodal status | Negative | 521 (41.4) | 571 (49.7) | <0.001 |
| | Positive | 736 (58.6) | 579 (50.3) | |
| Histologic grade | G1 | 29 (2.7) | 17 (1.6) | 0.25 |
| | G2 | 514 (47.9) | 501 (48.3) | |
| | G3 | 531 (49.4) | 519 (50.0) | |
| | Unknown | 183 | 113 | |
| Nuclear grade | G1 | 26 (2.9) | 19 (1.7) | 0.24 |
| | G2 | 436 (47.8) | 523 (47.9) | |
| | G3 | 450 (49.3) | 549 (50.3) | |
| | Unknown | 345 | 59 | |
| LVI | Negative | 490 (52.4) | 605 (57.8) | 0.015 |
| | Positive | 445 (47.6) | 441 (42.2) | |
| | Unknown | 322 | 104 | |
| Hormone receptor status | Negative | 457 (38.0) | 499 (44.8) | 0.001 |
| | Positive[a] | 752 (62.0) | 616 (55.2) | |
| | Unknown | 48 | 35 | |
| HER-2 (IHC) status | Negative | 833 (68.9) | 824 (73.9) | 0.008 |
| | Positive[b] | 376 (31.1) | 291 (26.1) | |
| | Unknown | 48 | 35 | |
| Subtype | HR+/HER-2- | 570 (47.1) | 506 (45.4) | <0.001 |
| | HR+/HER-2+ | 182 (15.1) | 110 (9.9) | |
| | HR-/HER-2+ | 194 (16.0) | 181 (16.2) | |
| | HR-/HER-2- | 263 (21.8) | 318 (28.5) | |
| | Unknown | 48 | 35 | |
| Radiotherapy | Yes | 508 (40.8) | 331 (29.3) | <0.001 |
| | No | 736 (59.2) | 799 (70.7) | |
| | Unknown | 13 | 20 | |
| Chemotherapy | Yes | 257 (20.7) | 302 (26.6) | 0.001 |
| | No | 985 (79.3) | 832 (73.4) | |
| | Unknown | 15 | 16 | |

(*Continued*)

**Table 2.** (Continued)

| Factors | | 2000–2007 (N = 1,257) | 2008–2013 (N = 1,150) | p-value |
|---|---|---|---|---|
| | | N (%) | N (%) | |
| Hormonal therapy | Yes | 434 (35.1) | 501 (44.5) | <0.001 |
| | No | 801 (64.9) | 624 (55.5) | |
| | Unknown | 22 | 25 | |
| Chemotherapy agent | None | 257 (22.5) | 296 (31.9) | <0.001 |
| | CMF | 47 (4.1) | 5 (0.5) | |
| | Anthracyclin-based | 436 (38.1) | 236 (25.5) | |
| | Taxane-based | 384 (33.6) | 373 (40.2) | |
| | Others | 19 (1.7) | 17 (1.8) | |
| | Unknown | 114 | 223 | |
| Hormonal therapy agent | None | 441 (35.8) | 503 (44.7) | <0.001 |
| | AI | 69 (5.6) | 122 (10.8) | |
| | SERM | 657 (53.4) | 404 (35.9) | |
| | SERM+LHRH analog | 64 (5.2) | 96 (8.5) | |
| | Unknown | 26 | 25 | |
| Chemotherapy after recurrence | Yes | 624 (61.9) | 526 (49.6) | <0.001 |
| | No | 384 (38.1) | 535 (50.4) | |
| | Unknown | 249 | 86 | |
| Hormonal therapy after recurrence | Yes | 536 (53.2) | 503 (47.4) | 0.009 |
| | No | 472 (46.8) | 558 (52.6) | |
| | Unknown | 249 | 86 | |
| Anti-targeted therapy after recurrence | Yes | 185(18.4) | 257 (24.2) | 0.001 |
| | No | 823 (81.6) | 804 (75.8) | |
| | Unknown | 249 | 86 | |

BCS, breast conserving surgery; LVI, lymphovascular invasion; [a]estrogen receptor-positive or progesterone receptor-positive; HER-2, human epidermal growth factor receptor-2; IHC, immunohistochemistry; [b]IHC 3+; CMF, cyclophosphamide, methotrexate, fluorouracil; AI, aromatase inhibitor; SERM, Selective estrogen receptor modulator; LHRH, luteinizing hormone-releasing hormone.

## Survival

The median follow-up duration from the time of relapse was 30.6 months (range, 0–223.4). During follow-up, a total of 1,391 deaths occurred, of which 1,348 (96.5%) were related to breast cancer. The 5-year rates of OS and SAR were 66.9% and 48.1%, respectively. The median survival duration after recurrence significantly increased from 38.0 months in period I to 49.7 months in period II (p < 0.001). In contrast, the increase in the median survival duration from period I (97.5 months) to period II (114.4 months) was not statistically significant (p = 0.092; Fig 1).

We analyzed the survival outcomes according to cancer subtypes to examine their influences on the improvements in survival outcomes over time (Figs 2 and 3). The SAR was

**Table 3. Distribution of the type of recurrence according to the year of recurrence.**

| Type of recurrence | 2000–2007 (N = 1,257) | 2008–2013 (N = 1,150) | Total (N = 2,407) |
|---|---|---|---|
| | N (%) | N (%) | N (%) |
| Loco-regional recurrence | 302 (24,0) | 404 (35.0) | 706 (29.0) |
| Systemic recurrence | 955 (76.0) | 746 (65.0) | 1701 (71.0) |

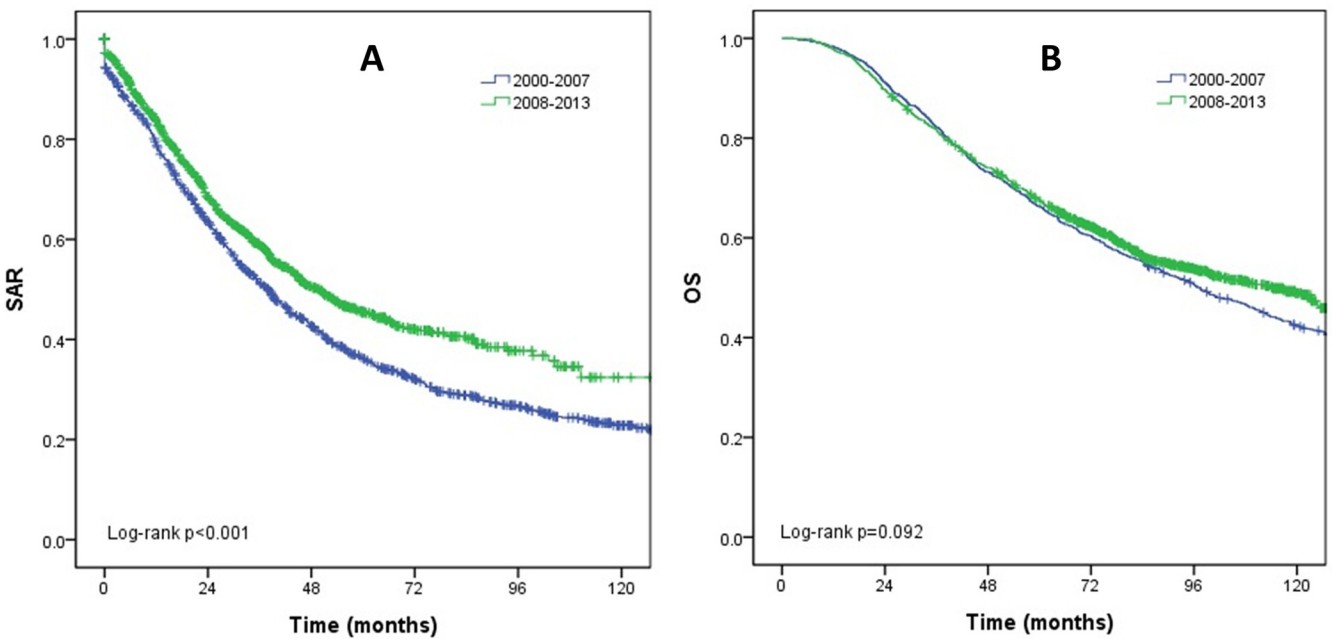

**Fig 1. Chronological changes in the survival rates of patients with recurred primary breast cancer.** (A) Survival after recurrence (SAR). (B) Overall survival (OS).

improved in the HR+/HER2- subtype (53.7 to 79.6 months; $p < 0.001$), HR+/HER2+ subtype (36.4 to 66.6 months; $p < 0.001$), and HR-/HER2+ subtype (24.7 to 49.1 months; $p < 0.001$). In contrast, the SAR of HR-/HER2- subtype was not improved ($p = 0.139$). On the other hand, the OS was improved only in the HR-/HER2+ subtype (55.6 to 95.9 months; $p < 0.001$) and not in other subtypes (HR+/HER2-; $p = 0.122$, HR+/HER2+; $p = 0.177$, HR-/HER2-; $p = 0.977$).

We performed multivariate Cox proportional hazards regression analyses to identify the factors influencing SAR (Table 4) and OS (Table 5). In the HR+/HER2+ subtype, the year of diagnosis was significantly associated with SAR (HR 0.58, 95% CI 0.37–0.9, $p = 0.015$) but not with OS (HR 0.77, 95% CI 0.49–1.21, $p = 0.258$). Older age at diagnosis was significantly associated with SAR in the HR+/HER2- subtype only (HR 2.18, 95% CI 1.56–3.05, $p < 0.001$). High T and N stages were associated with poorer survival. In HR-/HER2- patients, the tumor biology was significantly associated with survival, with histologic grade being significantly associated with OS (HR 1.34, 95% CI 1.06–1.71, $p = 0.014$) and LVI being significantly associated with SAR (HR 1.38, 95% CI 1.09–1.75, $p = 0.007$) and OS (HR 1.43, 95% CI 1.13–1.81, $p = 0.003$). Treatment after recurrence was generally significantly associated with survival. Hormonal therapy after recurrence was associated with better SAR and OS in HR+/HER2- patients. Likewise, target therapy after recurrence significantly increased the OS in all subtypes. The univariate Cox analysis was also performed and the results are provided in the S1–S4 Tables.

## Discussion

To our knowledge, this is the first study to compare recurred breast cancer outcomes, stratified by subtypes, over two distinct time periods. Our retrospective chronological study demonstrated changes in the survival rates of recurred breast cancer in Korean women in period I and period II. The SAR significantly improved between period I and period II, while the OS

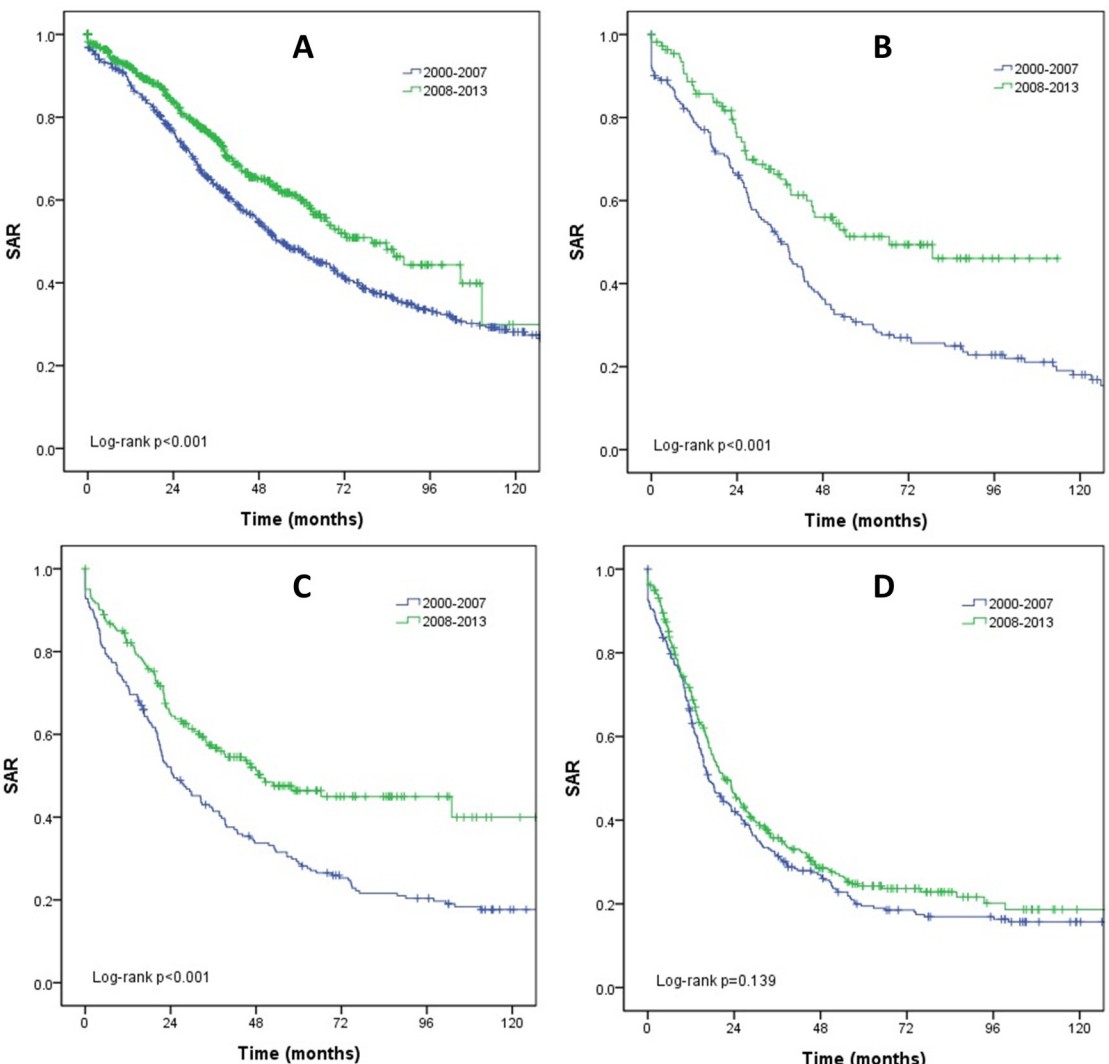

**Fig 2. Survival after recurrence (SAR) according to subtypes.** (A) HR+/HER2-. (B) HR+/HER2+. (C) HR-/HER2+. (D) HR-/
HER2-.

did not show a significant difference between the two periods. Similar to our result, a previous study performed in the same institution during the former period did not show a significant improvement in OS over time [10]. The previous study attributed the lack of significant OS differences to lead-time bias; however, we assume that the reason for this discrepancy may stem from the difference in treatment regimens depending on the cancer subtype. Therefore, we analyzed the survival outcomes according to subtypes and found that SAR was improved in other than HR-/HER2- subtype. Meanwhile, the OS was improved only in the HR-/HER2
+ subtype.

There are some factors that can influence the changes in survival rate in breast cancer patients who have recurred diseases, including advancements in adjuvant therapy such as hormonal and target therapy. The fact that the ratio of HR-positive and HER2-positive patients among relapsed patients was reduced (Table 2) may be regarded as possible evidence for this explanation.

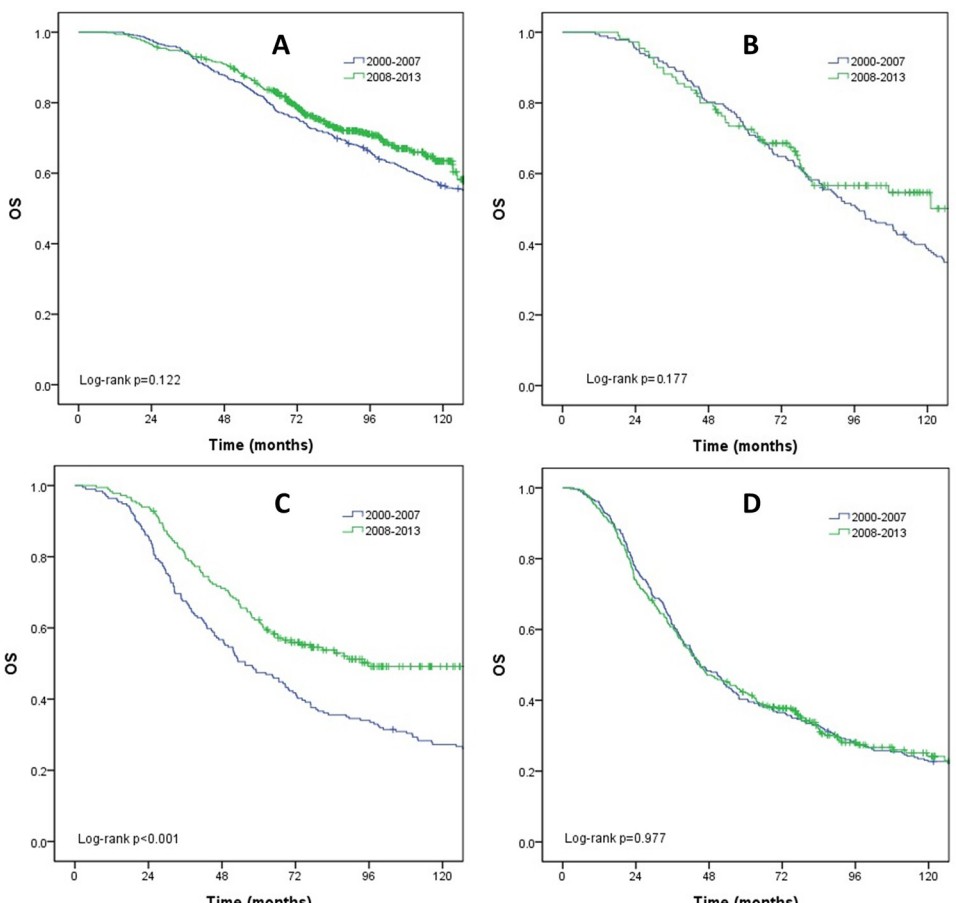

**Fig 3. Overall survival (OS) according to subtypes.** (A) HR+/HER2-. (B) HR+/HER2+. (C) HR-/HER2+. (D) HR-/HER2-.

Our findings suggest that advances in anti-hormone therapy and target therapy have contributed to improvements in the survival rate of breast cancer patients with recurrence. Initially, tamoxifen was mainly administered to HR-positive breast cancer patients. Aromatase inhibitors and LHRH agonists have been widely used since around 2003 [10], and the treatment of HER2-positive patients has advanced significantly. The use of trastuzumab for breast cancer patients was introduced in Korea around 2010. This also explains why OS was improved between period I and period II only in HER2-positive patients.

In our present study, the type of recurrence was analyzed between two periods. The relative ratio of systemic recurrence to loco-regional recurrence decreased. This result highlights the significant advancements in systemic therapy (e.g., anti-hormone therapy and target therapy) compared with local therapy (e.g., surgical technique or radiation therapy) during this period.

In multivariate analyses, age at diagnosis was associated with SAR and OS only in the HR-positive/HER2-negative subtype (Tables 4 and 5). In other subtypes, age was not an independent prognostic factor for survival after recurrence (Table 4). Several studies reported that young women have a high recurrence rate of breast cancer and poor prognosis [11–13]. The reason is considered to be due to poor clinicopathological features such as a low prevalence of luminal type and a relatively high prevalence of HER2 or TN types [14,15]. However, there is debate about whether age itself is an independent prognostic factor [16]. In this study, when

**Table 4. Multivariate Cox analysis for survival after recurrence (SAR).**

| Factors | HR+/HER2- | | HR+/HER2+ | | HR-/HER2+ | | HR-/HER2- | |
|---|---|---|---|---|---|---|---|---|
| | Hazard ratio | 95% CI | Hazard ratio | 95% CI | Hazard ratio | 95% CI | Hazard ratio | 95% CI |
| Year of diagnosis | | | | | | | | |
| 2000–2007 | 1.00 | Ref. | 1.00 | Ref. | 1.00 | Ref. | 1.00 | Ref. |
| 2008–2013 | 0.99 | 0.79–1.23 | **0.58** | **0.37–0.90** | 0.78 | 0.57–1.07 | 0.99 | 0.79–1.23 |
| Age at diagnosis (y) | | | | | | | | |
| 35–50 | 1.00 | Ref. | 1.00 | Ref. | 1.00 | Ref. | 1.00 | Ref. |
| <35 | 1.31 | 0.95–1.80 | 0.60 | 0.34–1.04 | 0.85 | 0.53–1.37 | 0.81 | 0.58–1.15 |
| >50 | **2.18** | **1.56–3.05** | 0.60 | 0.32–1.13 | 0.94 | 0.58–1.52 | 1.00 | 0.69–1.44 |
| T stage | | | | | | | | |
| T1 | 1.00 | Ref. | 1.00 | Ref. | 1.00 | Ref. | 1.00 | Ref. |
| T2 | 1.24 | 0.97–1.59 | 1.18 | 0.73–1.89 | **1.87** | **1.27–2.77** | 1.20 | 0.92–1.57 |
| T3 | **2.06** | **1.43–2.95** | 0.78 | 0.39–1.58 | **1.88** | **1.04–3.39** | **1.83** | **1.24–2.69** |
| T4 | **3.51** | **2.03–6.10** | **3.89** | **1.74–8.69** | **2.94** | **1.56–5.53** | 1.56 | 0.96–2.53 |
| Nodal stage | | | | | | | | |
| Negative | 1.00 | Ref. | 1.00 | Ref. | 1.00 | Ref. | 1.00 | Ref. |
| Positive | **1.66** | **1.29–2.14** | **2.10** | **1.32–3.36** | **1.97** | **1.34–2.91** | 1.28 | 1.00–1.64 |
| Histologic grade | | | | | | | | |
| G1 | 1.00 | Ref. | 1.00 | Ref. | 1.00 | Ref. | - | - |
| G2 | 2.42 | 0.76–7.71 | 1.62 | 0.54–4.87 | | | 1.00 | Ref. |
| G3 | **3.24** | **1.01–10.45** | 1.52 | 0.49–4.66 | | | 1.23 | 0.97–1.57 |
| LVI | | | | | | | | |
| No | 1.00 | Ref. | 1.00 | Ref. | 1.00 | Ref. | 1.00 | Ref. |
| Yes | 0.91 | 0.73–1.13 | 0.82 | 0.54–1.24 | 1.16 | 0.84–1.60 | **1.38** | **1.09–1.75** |
| Breast surgery | | | | | | | | |
| BCS | 1.00 | Ref. | 1.00 | Ref. | 1.00 | Ref. | 1.00 | Ref. |
| TM | 1.26 | 0.99–1.60 | 1.12 | 0.70–1.78 | 1.15 | 0.80–1.65 | 1.08 | 0.85–1.37 |
| Chemotherapy after recurrence | | | | | | | | |
| No | 1.00 | Ref. | 1.00 | Ref | 1.00 | Ref. | 1.00 | Ref. |
| Yes | **2.74** | **2.17–3.47** | **3.47** | **1.93–6.22** | **1.85** | **1.20–2.86** | **1.87** | **1.42–2.45** |
| Hormonal therapy after recurrence | | | | | | | | |
| No | 1.00 | Ref. | 1.00 | Ref. | 1.00 | Ref. | 1.00 | Ref. |
| Yes | **0.59** | **0.44–0.79** | 0.66 | 0.43–1.02 | 0.67 | 0.42–1.08 | 0.73 | 0.50–1.05 |
| Anti-targeted therapy after recurrence | | | | | | | | |
| No | 1.00 | Ref. | 1.00 | Ref. | 1.00 | Ref. | 1.00 | Ref. |
| Yes | 0.76 | 0.53–1.09 | **0.58** | **0.35–0.95** | **0.62** | **0.42–0.90** | 0.64 | 0.41–1.02 |

HR: Hormone receptor; HER2: Human epidermal growth factor receptor-2; LVI, lymphovascular invasion; BCS: Breast conserving surgery; TM: Total mastectomy

* The significant HRs (0.95 CI) are shown in bold.

HR+/HER2- subtype breast cancer patients with recurrence were analyzed, it was found that patients under 50 years of age had a higher survival rate after recurrence than did older patients. This may seem contradictory to the knowledge so far; however, this result should be interpreted with caution because only HR+/HER2- subtype patients were analyzed and they had received additional treatment after relapse. The reason for this result is that we implement active ovarian suppression therapy such as oophorectomy and LHRH agonist after relapse in premenopausal patients.

**Table 5. Multivariate Cox analysis for overall survivial (OS).**

| Factors | HR+/HER2- | | HR+/HER2+ | | HR-/HER2+ | | HR-/HER2- | |
|---|---|---|---|---|---|---|---|---|
| | Hazard ratio | 95% CI | Hazard ratio | 95% CI | Hazard ratio | 95% CI | Hazard ratio | 95% CI |
| Year of diagnosis | | | | | | | | |
| 2000–2007 | 1.00 | Ref. | 1.00 | Ref. | 1.00 | Ref. | 1.00 | Ref. |
| 2008–2013 | 1.03 | 0.82–1.20 | 0.77 | 0.49–1.21 | 0.86 | 0.63–1.19 | 1.13 | 0.90–1.42 |
| Age at diagnosis (y) | | | | | | | | |
| 35–50 | 1.00 | Ref. | 1.00 | Ref. | 1.00 | Ref. | 1.00 | Ref. |
| <35 | 1.09 | 0.79–1.51 | 0.71 | 0.41–1.22 | 0.79 | 0.55–1.43 | **0.67** | **0.47–0.95** |
| >50 | **1.61** | **1.15–2.24** | 0.62 | 0.33–1.15 | 0.90 | 0.55–1.46 | 0.82 | 0.56–1.19 |
| T stage | | | | | | | | |
| T1 | 1.00 | Ref. | 1.00 | Ref. | 1.00 | Ref. | 1.00 | Ref. |
| T2 | 1.22 | 0.96–1.56 | 1.46 | 0.91–2.34 | **2.03** | **1.37–3.01** | **1.33** | **1.02–1.74** |
| T3 | **2.29** | **1.59–3.30** | 1.11 | 0.55–2.22 | **2.01** | **1.10–3.65** | **2.15** | **1.47–3.14** |
| T4 | **4.77** | **2.73–8.35** | **7.97** | **3.57–17.78** | **3.76** | **1.99–7.09** | **2.19** | **1.35–3.55** |
| Nodal stage | | | | | | | | |
| Negative | 1.00 | Ref. | 1.00 | Ref. | 1.00 | Ref. | 1.00 | Ref. |
| Positive | **1.79** | **1.39–2.30** | **2.23** | **1.38–3.58** | **1.89** | **1.29–2.77** | 1.27 | 0.99–1.62 |
| Histologic grade | | | | | | | | |
| G1 | 1.00 | Ref. | 1.00 | Ref. | 1.00 | Ref. | - | - |
| G2 | **3.63** | **1.13–11.64** | 1.60 | 0.53–4.85 | | | 1.00 | Ref |
| G3 | **5.18** | **1.60–16.82** | 1.86 | 0.60–5.76 | | | **1.34** | **1.06–1.71** |
| LVI | | | | | | | | |
| No | 1.00 | Ref. | 1.00 | Ref. | 1.00 | Ref. | 1.00 | Ref. |
| Yes | 0.89 | 0.72–1.10 | 0.86 | 0.57–1.31 | 1.17 | 0.85–1.62 | **1.43** | **1.13–1.81** |
| Breast surgery | | | | | | | | |
| BCS | 1.00 | Ref. | 1.00 | Ref. | 1.00 | Ref. | 1.00 | Ref. |
| TM | 1.26 | 1.00–1.60 | 0.98 | 0.62–1.56 | 1.34 | 0.93–1.93 | 1.21 | 0.95–1.53 |
| Chemotherapy after recurrence | | | | | | | | |
| No | 1.00 | Ref. | 1.00 | Ref. | 1.00 | Ref. | 1.00 | Ref. |
| Yes | **2.93** | **2.33–3.70** | **5.19** | **2.80–9.60** | **1.86** | **1.19–2.89** | **2.08** | **1.59–2.74** |
| Hormonal therapy after recurrence | | | | | | | | |
| No | 1.00 | Ref. | 1.00 | Ref. | 1.00 | Ref. | 1.00 | Ref. |
| Yes | **0.69** | **0.51–0.92** | 0.75 | 0.49–1.16 | 0.63 | 0.39–1.01 | **0.66** | **0.45–0.95** |
| Anti-targeted therapy after recurrence | | | | | | | | |
| No | 1.00 | Ref. | 1.00 | Ref. | 1.00 | Ref. | 1.00 | Ref. |
| Yes | **0.66** | **0.46–0.94** | **0.51** | **0.30–0.86** | **0.60** | **0.41–0.88** | **0.56** | **0.35–0.88** |

HR: Hormone receptor; HER2: Human epidermal growth factor receptor-2; LVI, lymphovascular invasion; BCS: Breast conserving surgery; TM: Total mastectomy.

* The significant HRs (0.95 CI) are shown in bold.

In multivariate analysis, higher tumor stages (i.e., large tumor size and axillary lymph node metastasis) were significantly associated with poor prognosis in most subtypes; however, in triple-negative subtype patients, tumor stages failed to show significant correlations. While the reason for the relatively weak associations of tumor stages with survival outcomes in triple-negative patients is unclear, this may be at least partly due to the poor prognosis of early-stage triple-negative breast cancer [17].

In the analysis of treatment after relapse, chemotherapy after recurrence was significantly associated with poor survival outcomes in all subtypes (Tables 4 and 5). Administration of

adjuvant chemotherapy in initial cancer treatment was reported to be associated with poor prognosis after recurrence [18]. Goldhirsch et al. attributed this association to the fact that the reason for the chemotherapy was related to the poor standard risk factor that the disease had [19]. According to the current treatment protocol, chemotherapy after recurrence in all subtypes is closely related to systemic metastasis. Targeted therapy after recurrence was a treatment-related factor that improved survival in HER2-positive patients. This supports the results of previous studies that the development of trastuzumab made a significant difference in the improvement of the survival rate of relapsed breast cancer patients after surgery [20]. On the other hand, hormonal therapy after recurrence improved the survival rate in luminal A type but not significantly so in other subtypes.

In a previous study on the survival rate of relapsed patients in the United States, the year of relapse was an independent factor associated with the survival rate [21]. Meanwhile, in our multivariate analysis, after adjusting for prognostic factors that were previously dealt with, the year of diagnosis was not significantly associated with improved survival after recurrence. This may be due to the fact that in the modern era, early detection of breast cancers has increased due to wider screening and the tumor stages are lower at diagnosis; moreover, the development of chemotherapy such as taxanes as well as hormonal and target therapy may have also contributed to such results [22–24].

The findings of this study are subject to the following limitations. First, it is a retrospective study performed at a single center, which is prone to selection bias. For example, physicians might have prescribed more intensive treatments in patients considered to have a high risk of recurrence than those with a low risk. Second, the drugs specifically used for treatment need to be investigated in targeted studies to confirm our results that the advance in treatment was associated with improvements in survival in recurrent breast cancer patients. Nevertheless, this study is meaningful because these limitations reflect actual practice. Third, there were significant differences in some pathological variables between the two periods. However, we did not match the variables to generalize the results. Lastly, further research is needed to determine whether the associations found in the current study are causal.

In conclusion, the results of our current analysis suggest that while OS and SAR improved in recurred breast cancer patients over time, the improvements in survival outcomes were different in each subtype. As age at diagnosis, cancer stage (tumor size and lymph node status), and adjuvant therapy regimen after recurrence were significant prognostic factors for survival in relapsed patients, they may be helpful in planning adjuvant treatment strategies for each subtype.

## Supporting information

**S1 Fig. Kaplan–Meier curve and log-rank test using OS.**
(TIF)

**S2 Fig. Kaplan–Meier curve and log-rank test using SAR.**
(TIF)

**S1 Table. Univariate Cox analysis for survival after recurrence (HR+/HER2-).**
(DOCX)

**S2 Table. Univariate Cox analysis for survival after recurrence (HR+/HER2+).**
(DOCX)

**S3 Table. Univariate Cox analysis for survival after recurrence (HR-/HER2+).**
(DOCX)

**S4 Table. Univariate Cox analysis for survival after recurrence (HR-/HER2-).**
(DOCX)

**S1 Data.**
(XLSX)

## Author Contributions

**Conceptualization:** Young-jin Lee, Sae Byul Lee.

**Data curation:** Young-jin Lee.

**Formal analysis:** Young-jin Lee, Sae Byul Lee.

**Investigation:** Young-jin Lee, Sae Byul Lee.

**Methodology:** Young-jin Lee, Sae Byul Lee.

**Resources:** Young-jin Lee.

**Software:** Young-jin Lee.

**Supervision:** Tae-Kyung Yoo, Jisun Kim, Il Yong Chung, Beom Seok Ko, Hee Jeong Kim, Jong Won Lee, Byung Ho Son, Sei-Hyun Ahn, Sae Byul Lee.

**Validation:** Tae-Kyung Yoo, Jisun Kim, Il Yong Chung, Beom Seok Ko, Hee Jeong Kim, Jong Won Lee, Byung Ho Son, Sei-Hyun Ahn, Sae Byul Lee.

**Visualization:** Young-jin Lee.

**Writing – original draft:** Young-jin Lee.

**Writing – review & editing:** Young-jin Lee.

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
