## [Decision Letter · Decision Letter 0]

19 Sep 2022

PONE-D-22-18837Survival outcomes of breast cancer patients with recurrence after surgery according to period and subtypePLOS ONE

Dear Dr. Lee,

Thank you for submitting your manuscript to PLOS ONE. After careful consideration, we feel that it has merit but does not fully meet PLOS ONE’s publication criteria as it currently stands. Therefore, we invite you to submit a revised version of the manuscript that addresses the points raised during the review process.

We look forward to receiving your revised manuscript.

Kind regards,

Yasunori Sato

Academic Editor

PLOS ONE

Journal Requirements:

   "This study was supported by a grant (Elimination of Cancer Project Fund) from the Asan Cancer Institute of Asan Medical Center, Seoul (2017-1341)."

 "The authors received no specific funding for this work."

Reviewers' comments:

Reviewer's Responses to Questions

**Comments to the Author**

1. Is the manuscript technically sound, and do the data support the conclusions?

Reviewer #1: Partly

Reviewer #2: Partly

Reviewer #3: Yes

2. Has the statistical analysis been performed appropriately and rigorously? 

Reviewer #1: Yes

Reviewer #2: I Don't Know

Reviewer #3: Yes

3. Have the authors made all data underlying the findings in their manuscript fully available?

Reviewer #1: No

Reviewer #2: Yes

Reviewer #3: Yes

4. Is the manuscript presented in an intelligible fashion and written in standard English?

Reviewer #1: No

Reviewer #2: Yes

Reviewer #3: Yes

5. Review Comments to the Author

Reviewer #1: The manuscript titled ‘Survival outcomes of breast cancer patients with recurrence after surgery according to period and subtype’ aimed to compare survival outcomes in BC patients between two periods of time o identify the factors associated with post-recurrence survival and overall survival and changes over time in the duration of survival after recurrence. Following comments could be helpful for authors.

Minor comments:

- Don’t mention the definition of period1 and period2 multiple times (lines 21,25,55, 63, 169)! Mentioning once would be enough!

- Your sample size is 2,407, so report it in abstract and method as your sample size as your analysis is based on that.

- It seems that there is an abbreviation in the literature stands for post-recurrence survival (PRS).

Major comments:

- There is a lack of proper rationales in introduction part. Why this study is important?

- SAR should be defined from the time of recurrence to death/last follow up. OS is also defined time from diagnosis to death/last follow up time.

- What about eligibility criteria! Is your study restricted to age (e.g., 18-77), those who had not second recurrence or metastasis etc.

- If study aimed to see improvement in treatment in patients who had recurrence, the time of treatment is of importance. If so, the treatments for patients with recurrence should be considered as well! You don’t know is this the effect of treatments before recurrence or after the recurrence. This bias can affect the result. Were patients received the treatment after their recurrence? There is no information in the manuscript to support this! The treatments mentioned in table 2, are received by patients after recurrence?

- In table 2, comparison results showed that in some pathologic variables, there are significant differences between two periods (e.g., T stage, Nodal status, HR etc)! It seems that you might need to match comparison groups. The question is that are they comparable?

- Your sample size covers recurred patients (your title), right? while you have results based on overall survival which includes interval time from diagnosis to death/last follow up!

- Be clear on your sample size! It looks Analyses were performed based on recurred sample! If you are analyzing 17,776 patients, mention the sample of analysis in table 4 and 5.

- Why didn’t you use trend analysis to see the changes over time?

Reviewer #2: Sept 5, 2022

Title: Survival outcomes of breast cancer patients with recurrence after surgery according to period and subtype

This paper presents a study to analyze and compare the survival rates of recurrent breast cancer patients in Korea between two periods (2000–2007; 2008–2013) and to identify the factors associated with outcomes and changes over time in the duration of survival after recurrence. However, there are questions that limit my enthusiasm of the paper, as outlined below.

1. Authors include the patients with unknown clinical variables e.g., stage, histology, etc. Why not removing these patients? If for any clinical and biological reason, this group named as “unknown” is important and would like to assess the effect of that in analyses, still not correct analysis was applied. Authors considered chi-squared method to do the association which will have low power in this scenario, and the non-parametric Fisher exact test is the correct method to be applied.

2. Authors did mention that multivariate Cox model was applied, and I couldn’t follow in which step this method was applied. I already saw the findings using univariate Cox model.

3. Please keep the HR estimate along with p value to report findings.

4. Table 3: Please add appropriate test to assess the association between type of recurrence and year of recurrence.

5. Figures 2 and 3: How about adding supplementary figures to show the KM and log-rank test using OS across all four subtypes and the same test for SAR.

6. Tables (e.g., 4 and 5), for example for some categorical variables e.g., age, T stage, etc., we have the p values and other statistics for each category, while a p value for that given variable (e.g., age, T stage). I couldn’t follow how the p value was computed.

7. In addition, authors only considered clinical data, while adding molecular data along with clinical data to assess the association with OS or SAR can make the paper much more interesting along with this fact it can be fit better for PLOS ONE journal requirement.

Reviewer #3: 1. Using anti-hormonal therapy instead of hormonal therapy is not common. Even though It is not incorrect, it is suggested to change it.

2. Table 3: According to data in table 3, no patient had a locoregional and systematic recurrence. While the co-incidence of outcomes is possible.

3. Line 101: You have presented that the follow-up duration from the time of relapse ranged between 0–223.4 months. Since the data is pertaining to 2000-2013 and they have been recruited into the database from 2017 to 2021, one expects that their follow-up time is not zero. Would you please explain more about this data?

4. Please define the statistical tests which were used in the analysis of data presented in lines 131-135.

5. Line 134-135: In this sentence, “… the median 5-year OS rate 135 from the period I (97.5 months) to period II (114.4 months)”, you have not presented any survival rate, and the sentence needs an edition. Those values are the median survival durations.

6. Tables 4 and 5 are huge boxes of data and are hard to use. The p-values can be deleted, and the significant HR (0.95 CI)s can be bolded.

7. Line 191: regarding this sentence and the following explanation, “In multivariate analyses, age at diagnosis was independently associated with…” we should notice that independent association of variables is studied in univariate analysis. In multivariate analysis, the effect of each variable is dependent on the effect of other included variables in the regression model.

8. Line 208: “chemotherapy after recurrence was significantly associated with survival outcomes…”. It is not clear whether it was associated with better or worse survival?

6. PLOS authors have the option to publish the peer review history of their article (what does this mean?). If published, this will include your full peer review and any attached files.

Reviewer #1: No

Reviewer #2: No

Reviewer #3: **Yes: **Shahpar Haghighat

---

## [Author Response · Author response to Decision Letter 0]

6 Jan 2023

Response to Reviewer #1

Thank you for your review of our paper. We have answered each of your points below.

Comment 1: Don’t mention the definition of period1 and period2 multiple times (lines 21,25,55, 63, 169)! Mentioning once would be enough! 

Response: Corrected as per the reviewer's comment. We mentioned the definition once. (lines 25,56,63,169)

Comment 2: Your sample size is 2,407, so report it in abstract and method as your sample size as your analysis is based on that.

Response: Corrected as per the reviewer's comment. (lines 24,62)

“We retrospectively analyzed 2,407 patients who had recurrent breast cancer with treated between January 2000 and December 2013 and divided them into two periods according to the year of recurrence.” (line 24)

“Among them, we selected 2,407 patients who experienced recurrence before December 31, 2020.” (line 62)

Comment 3: It seems that there is an abbreviation in the literature stands for post-recurrence survival (PRS).

Response: Corrected as per the reviewer's comment. (lines 57)

“By doing so, we tried to identify the factors associated with overall and post-recurrence survival”. (line 57)

Comment 4: There is a lack of proper rationales in introduction part. Why this study is important? 

Response: The introduction part has been reinforced as the reviewer’s comment. 

“Therefore, it seems important to analyze the impact of the development of adjuvant therapy for each subtype over time on the prognosis of relapsed patients.” (line 51)

Comment 5: SAR should be defined from the time of recurrence to death/last follow up. OS is also defined time from diagnosis to death/last follow up time. 

Response: Corrected as per the reviewer's comment. 

“Overall survival (OS) was defined as the time from surgery to death/last follow up. SAR was defined as the time from recurrence to death/last follow up by referring to the Korean registry cause-of-death records.” (line 70)

Comment 6: What about eligibility criteria! Is your study restricted to age (e.g., 18-77), those who had not second recurrence or metastasis etc.

Response: Thank you for your comment. The age of the subjects of this study is 21-88 years. Among them, 14 patients over 77 years of age and only 7 patients over 80 years of age are expected to have no significant effect on the overall outcome. For more than 2 recurrences, there are not enough data at this time, so we plan to analyze them further later.

Comment 7: If study aimed to see improvement in treatment in patients who had recurrence, the time of treatment is of importance. If so, the treatments for patients with recurrence should be considered as well! You don’t know is this the effect of treatments before recurrence or after the recurrence. This bias can affect the result. Were patients received the treatment after their recurrence? There is no information in the manuscript to support this! The treatments mentioned in table 2, are received by patients after recurrence?

Response: Among the treatments mentioned in Table 2, if 'after recurrence' is appended to the name of the treatment, it means the treatment applied after the recurrence, and otherwise means the adjuvant therapy received during the first treatment. For this, we have added content to the method. Treatment after recurrence was used in Cox regression analysis.

All information about the patients and diseases was retrieved from the retrospectively collected database, including age, clinical manifestations, clinical and pathologic staging according to the American Joint Committee on Cancer classification, pathologic data, surgical methods, types of adjuvant therapy received during the first treatment, types of post-relapse adjuvant therapy which marked 'after recurrence', type of recurrence, and follow-up period. (line 68)

Comment 8: In table 2, comparison results showed that in some pathologic variables, there are significant differences between two periods (e.g., T stage, Nodal status, HR etc)! It seems that you might need to match comparison groups. The question is that are they comparable?

Response: If we match the variables, the results cannot be generalized. That's why we don't match comparison groups. It is a limitation of our study and has been added to the discussion section. And multivariate Cox model was used to minimize the influence of other variables.

“Third, there are significant differences in some pathologic variables between two periods. However, we do not match the variables to generalize the results.” (line 234)

Comment 9: Your sample size covers recurred patients (your title), right? while you have results based on overall survival which includes interval time from diagnosis to death/last follow up!

Response: Yes, our sample size covers recurred patients. Our results based on overall survival as well as survival after recurrence.

Comment 10: Be clear on your sample size! It looks Analyses were performed based on recurred sample! If you are analyzing 17,776 patients, mention the sample of analysis in table 4 and 5.

Response: We analyzed 2,407 patients. The manuscript have been corrected in previous comments.

Comment 11: Why didn’t you use trend analysis to see the changes over time?

Response: In this study, trend analysis was not performed because we thought that the focus was on the difference between the two periods rather than the serial change.

Response to Reviewer #2

Thank you for reviewing our manuscript. Our answers to your queries are as follows.

Comment 1: Authors include the patients with unknown clinical variables e.g., stage, histology, etc. Why not removing these patients? If for any clinical and biological reason, this group named as “unknown” is important and would like to assess the effect of that in analyses, still not correct analysis was applied. Authors considered chi-squared method to do the association which will have low power in this scenario, and the non-parametric Fisher exact test is the correct method to be applied.

Response: The unknown groups of each variables are removed when we do this uni/multivariate Cox regression analysis. The methods part has been reinforced to clarify this.

The unknown groups of each variables are removed before Cox analysis proceeded. (line 100)

Comment 2: Authors did mention that multivariate Cox model was applied, and I couldn’t follow in which step this method was applied. I already saw the findings using univariate Cox model.

Response: We did univariate Cox regression analysis to determine how the factors impact on survival. And then we did multivariate Cox regression analysis to remove the interaction of multiple, potentially interacting covariates. We selected as variables the periods, age at diagnosis, adjuvant therapy after recurrence, and pathologic variables that we are thought to affect survival with other covariates.

Comment 3: Please keep the HR estimate along with p value to report findings.

Response: Corrected as per the reviewer's comment. (lines 157-164) 

Comment 4: Table 3: Please add appropriate test to assess the association between type of recurrence and year of recurrence.

Response: We added chi-squared tests to assess it.

There was significant differences in type of recurrence according to time period (p < 0.001). (line 128)

Comment 5: Figures 2 and 3: How about adding supplementary figures to show the KM and log-rank test using OS across all four subtypes and the same test for SAR.

Response: We added supplementary figures to show the KM and log-rank test using OS and SAR across all four subtypes.

Comment 6: Tables (e.g., 4 and 5), for example for some categorical variables e.g., age, T stage, etc., we have the p values and other statistics for each category, while a p value for that given variable (e.g., age, T stage). I couldn’t follow how the p value was computed.

Response: The statistical analysis section of method part has been reinforced as the reviewer’s comment.

A multivariate Cox regression analysis with a backward elimination method was used to estimate the hazard ratios and p values and to identify independent prognostic factors. (line 98)

Comment 7: In addition, authors only considered clinical data, while adding molecular data along with clinical data to assess the association with OS or SAR can make the paper much more interesting along with this fact it can be fit better for PLOS ONE journal requirement.

Response: Thank you for your considerate comment. For now, collecting molecular data is a challenging problem. As a follow-up study, we will consider the analysis of molecular data. 

Response to Reviewer #3

Thank you for your review of our paper. We have answered each of your points below.

Comment 1: Using anti-hormonal therapy instead of hormonal therapy is not common. Even though It is not incorrect, it is suggested to change it.

Response: We changed the terminology as the reviewer’s comment. 

Comment 2: Table 3: According to data in table 3, no patient had a locoregional and systematic recurrence. While the co-incidence of outcomes is possible. 

Response: There are 615 patients who had locoregional and systemic recurrence at the same time. We included these patients in the ‘systemic recurrence’ category. 

Comment 3: Line 101: You have presented that the follow-up duration from the time of relapse ranged between 0–223.4 months. Since the data is pertaining to 2000-2013 and they have been recruited into the database from 2017 to 2021, one expects that their follow-up time is not zero. Would you please explain more about this data? 

Response: There are some patients who are lost to follow up. Their follow-up duration from the time of relapse was counted as zero.

Comment 4: Please define the statistical tests which were used in the analysis of data presented in lines 131-135.

Response: The median survival duration is the duration at which the probability of survival equals 50%. Survival curves were generated using the Kaplan–Meier method, and the significance was verified using the log-rank test.

Comment 5: Line 134-135: In this sentence, “… the median 5-year OS rate 135 from the period I (97.5 months) to period II (114.4 months)”, you have not presented any survival rate, and the sentence needs an edition. Those values are the median survival durations.

Response: Corrected as per the reviewer's comment.

The median survival duration after recurrence significantly increased from 38.0 months in period I to 49.7 months in period II (p < 0.001). In contrast, the increase in the median survival duration from period I (97.5 months) to period II (114.4 months) was not statistically significant (p = 0.092; Fig 1). (line 137-140)

Comment 6: Tables 4 and 5 are huge boxes of data and are hard to use. The p-values can be deleted, and the significant HR (0.95 CI)s can be bolded.

Response: We revised Table 4 and 5 as the reviewer’s comment.

Comment 7: Line 191: regarding this sentence and the following explanation, “In multivariate analyses, age at diagnosis was independently associated with…” we should notice that independent association of variables is studied in univariate analysis. In multivariate analysis, the effect of each variable is dependent on the effect of other included variables in the regression model.

Response: Thank you for your considerate comment. We reviewed this part again and revised the manuscript.

“In multivariate analyses, age at diagnosis was associated with SAR and OS only in HR-positive/HER2-negative subtype (Table 4, 5).” (line 196)

Comment 8: Line 208: “chemotherapy after recurrence was significantly associated with survival outcomes…”. It is not clear whether it was associated with better or worse survival?.

Response: Corrected as per the reviewer's comment.

“In the analysis of treatment after relapse, chemotherapy after recurrence was significantly associated with worse survival outcomes in all subtypes (Table 4, 5)” (line 212)

Thank you again for reviewing our manuscript in detail and providing helpful comments. We hope that our responses and the corresponding revisions are satisfactory.

---

## [Decision Letter · Decision Letter 1]

3 Apr 2023

Survival outcomes of breast cancer patients with recurrence after surgery according to period and subtype

PONE-D-22-18837R1

Dear Dr. Lee,

We’re pleased to inform you that your manuscript has been judged scientifically suitable for publication and will be formally accepted for publication once it meets all outstanding technical requirements.

Kind regards,

Yasunori Sato

Academic Editor

PLOS ONE

Reviewers' comments:

Reviewer's Responses to Questions

**Comments to the Author**

1. If the authors have adequately addressed your comments raised in a previous round of review and you feel that this manuscript is now acceptable for publication, you may indicate that here to bypass the “Comments to the Author” section, enter your conflict of interest statement in the “Confidential to Editor” section, and submit your "Accept" recommendation.

Reviewer #2: All comments have been addressed

Reviewer #3: All comments have been addressed

2. Is the manuscript technically sound, and do the data support the conclusions?

Reviewer #2: Yes

Reviewer #3: No

3. Has the statistical analysis been performed appropriately and rigorously? 

Reviewer #2: Yes

Reviewer #3: Yes

4. Have the authors made all data underlying the findings in their manuscript fully available?

Reviewer #2: Yes

Reviewer #3: Yes

5. Is the manuscript presented in an intelligible fashion and written in standard English?

Reviewer #2: Yes

Reviewer #3: Yes

6. Review Comments to the Author

Reviewer #2: Title: Survival outcomes of breast cancer patients with recurrence after surgery according to period and subtype

Authors addressed all the comments. Thank you

Reviewer #3: (No Response)

7. PLOS authors have the option to publish the peer review history of their article (what does this mean?). If published, this will include your full peer review and any attached files.

Reviewer #2: No

Reviewer #3: **Yes: **Shahpar Haghighat

---

## [Editor Report · Acceptance letter]

12 Apr 2023

PONE-D-22-18837R1 

Survival outcomes of breast cancer patients with recurrence after surgery according to period and subtype 

Dear Dr. Lee:

I'm pleased to inform you that your manuscript has been deemed suitable for publication in PLOS ONE. Congratulations! Your manuscript is now with our production department. 

Kind regards, 

on behalf of

Dr. Yasunori Sato 

Academic Editor

PLOS ONE